# Impact of Chromosomal Context on Origin Selection and the Replication Program

**DOI:** 10.3390/genes13071244

**Published:** 2022-07-14

**Authors:** Lilian Lanteri, Anthony Perrot, Diane Schausi-Tiffoche, Pei-Yun Jenny Wu

**Affiliations:** 1Institute of Genetics and Development of Rennes, UMR6290, CNRS—University of Rennes, 35000 Rennes, France; lilian.lanteri@gmail.com (L.L.); anthony.perrot49@gmail.com (A.P.); diane.schausi@univ-rennes1.fr (D.S.-T.); 2Institute of Biochemistry and Cellular Genetics, UMR5095, CNRS—University of Bordeaux, 33077 Bordeaux, France

**Keywords:** DNA replication, replication program, chromosomal architecture, genome rearrangements, fission yeast

## Abstract

Eukaryotic DNA replication is regulated by conserved mechanisms that bring about a spatial and temporal organization in which distinct genomic domains are copied at characteristic times during S phase. Although this replication program has been closely linked with genome architecture, we still do not understand key aspects of how chromosomal context modulates the activity of replication origins. To address this question, we have exploited models that combine engineered genomic rearrangements with the unique replication programs of post-quiescence and pre-meiotic S phases. Our results demonstrate that large-scale inversions surprisingly do not affect cell proliferation and meiotic progression, despite inducing a restructuring of replication domains on each rearranged chromosome. Remarkably, these alterations in the organization of DNA replication are entirely due to changes in the positions of existing origins along the chromosome, as their efficiencies remain virtually unaffected genome wide. However, we identified striking alterations in origin firing proximal to the fusion points of each inversion, suggesting that the immediate chromosomal neighborhood of an origin is a crucial determinant of its activity. Interestingly, the impact of genome reorganization on replication initiation is highly comparable in the post-quiescent and pre-meiotic S phases, despite the differences in DNA metabolism in these two physiological states. Our findings therefore shed new light on how origin selection and the replication program are governed by chromosomal architecture.

## 1. Introduction

DNA replication is an essential process that is tightly controlled to ensure the duplication and transmission of the genetic material. The initiation of DNA synthesis occurs at sites called origins of replication and relies on the ordered recruitment of conserved replication factors. The number of origins identified in eukaryotic organisms ranges from a few hundred to tens of thousands [1,2,3], and the firing of multiple origins during each S phase promotes the completion of DNA replication prior to cell division. However, not all initiation sites are activated during each cell cycle, and origins in a given genome are not used equivalently [1,2,3]. Each origin is defined by two major characteristics: its timing of firing during S phase and its efficiency, or frequency of initiation, in a population of cells. Interestingly, these two parameters are correlated in yeast models, as efficient origins tend to replicate earlier in S phase compared to inefficient origins, which are generally late-replicating [4,5,6]. Altogether, the timings, efficiencies, and distribution of origins along the chromosomes give rise to a spatiotemporal pattern of DNA synthesis referred to as the replication program.

Despite the diversity in the sequences of origins identified in different organisms, a large body of work suggests that origin activity is governed by similar principles [1,2,3]. In particular, the sites of replication initiation have been associated with features of open chromatin [7]. This is further illustrated by their localization in nucleosome-depleted regions (NDR) [8,9,10] and the observation that lower nucleosome occupancy around an origin is linked with higher origin efficiency [11]. The acetylation status of origins has also been correlated with their activity and timing of firing [12,13]. For instance, in budding yeast, deletion of the *RPD3* histone deacetylase induces late origins to fire earlier, and tethering the histone acetyltransferase Gcn5 near late origins has a similar effect [14]. Moreover, a relationship between replication initiation and transcription has been described in multiple systems [1,15]. Indeed, euchromatin tends to replicate early in S phase, while heterochromatic regions are mostly late-replicating [16,17]. This coupling is also supported by the fact that transcription initiation and levels of gene expression have been associated with the timing and efficiency of origin usage [18,19,20]. The results of these studies thus suggest that the open chromatin state at transcriptionally active loci may favor the firing of origins in these regions.

However, the interplay between chromatin structure and replication is far from straightforward. For example, a significant portion of origins in HeLa cells mapped in one study lack the histone marks for open chromatin [21], and deacetylation of histone H4 has been shown to promote firing of several origins in budding yeast [22]. Related to these observations, heterochromatic loci such as the pericentromeric regions and mating-type cassettes in the fission yeast *Schizosaccharomyces pombe* are duplicated early in S phase [23,24]. Furthermore, few replication initiation events were detected in areas regions with high transcription rates in human cell lines [25], and the silencing of origin activity in the temporal transition region (TTR) of the mouse immunoglobulin heavy chain (*Igh*) was not relieved by insertion of a highly active transcription unit, despite inducing a local increase in marks of active chromatin [26]. Altogether, these findings highlight the complexity and multi-layered regulation of the processes that modulate origin activity.

Importantly, a conserved characteristic of the replication program in eukaryotes is the organization of the genome in distinct domains that are duplicated at specific times in S phase. In fact, the observation of such reproducible patterns of origin usage in population studies, as well as the detection of clusters of origins in single-molecule and single-cell experiments, provided evidence that chromosomal context may be integral to the regulation of DNA replication [5,27,28,29,30,31,32]. Neighboring origins were also shown to fire at similar times, and genome-wide chromosome conformation capture analyses revealed that origins that fire concomitantly tend to contact one another [15,33,34,35]. Notably, these approaches uncovered correlations between topologically associated domains (TADs) and domains of replication timing, both of which are established during early G1 [35]. Specific *cis*-acting features, known as Early Replication Control Elements (ERCEs), were subsequently shown to be responsible for promoting early replication and contributing to TAD structure [36]. These investigations therefore represented essential steps in our understanding of the central role of chromosomal architecture in the establishment of the replication program.

Although both the characteristics of origins and the organization of the chromosomes are involved in origin selection, the crosstalk between these two levels of regulation remains unclear. Intriguingly, studies that have assessed the activities of origins artificially integrated at ectopic loci suggested that individual initiation sites are subjected to modulation by genomic context. For example, in the *Igh* TTR of mouse ES cells, where origins are silenced, insertion of normally constitutively active origins led to very few replication initiation events at these new sites [26]. Position-dependent regulation of origin firing has also been observed in fission yeast: integration of a normally efficient origin in either efficient or inefficient genomic regions resulted in a maintenance or a reduction of its activity, respectively [37]. While these studies indicate that the sequence of an origin is not sufficient to determine its activity, it remains difficult in these assays to distinguish between the loss of local features that may define origin selection vs. the position of an origin within in the large-scale organization of the genome. In this context, analyses of the effects of chromosomal rearrangements have yielded intriguing insights. Indeed, chromosome-wide delays in replication timing and completion of DNA synthesis, as assayed by BrdU incorporation and mitotic spreads, were induced by a subset of inter-chromosomal rearrangements in a human fibrosarcoma cell line, HTD114 [38]. In addition, characterization of an extensively rearranged copy of human chromosome 21 in murine cells revealed that small fragments (<500 kb) lost timing control and were duplicated at the same time as their flanking sequences, while larger fragments (>1 Mb) maintained their replication timing regardless of genomic properties [39]. Interestingly, genome re-organization also occurs during evolution, providing an opportunity to assess potential accompanying changes in DNA replication. For instance, comparisons of mouse and human syntenic regions indicates an overall conservation of replication timing, despite numerous chromosomal rearrangements, but divergences were found at a subset of fusion points [40]. Such deviations were also observed when comparing *Lachancea* yeast, where correlations in replication timing between species were, on average, weaker around synteny breakpoints [41]. Collectively, these studies underscore the absence of a unifying model that describes how chromosomal context modulates DNA replication, and a number of critical questions remain unanswered. In particular, we still do not understand how the selection of origins for activation is shaped by the interplay between their individual characteristics, chromosomal positions, and genomic context.

To explore this question, we have taken advantage of the signature replication program used by fission yeast cells that are subjected to conditions of reduced nitrogen availability [42]. Upon nitrogen starvation, cells exit the division cycle and enter a quiescent state, and subsequent addition of a nitrogen source promotes the resumption of proliferation. During the first post-quiescent S phase, striking differences are observed in the efficiencies of distinct domains, with areas of high origin activity adjoining regions that contain few initiation sites. In addition, we have previously shown that diploid cells subjected to the same treatment but induced to enter meiosis at the time of cell cycle re-entry undergo pre-meiotic S phase with a replication program similar to that in post-quiescent cells. In the present study, we have used these models to probe how chromosomal context may shape origin selection in physiological situations that exhibit marked differences in chromosome biology. To this end, we engineered large genomic inversions that alter characterized efficiency domains and determined the consequences for replication initiation. Our results show that these gross chromosomal rearrangements surprisingly do not bring about defects in cell proliferation, cell cycle re-entry, or meiotic progression, although we observe a restructuring of DNA replication along the chromosomes. Remarkably, this is brought about solely by changes in origin positioning, as most loci maintain their individual efficiencies. This underlines the robustness of origin selection to large-scale changes in chromosomal organization. However, we identified specific differences in origin activity near the fusion points of each rearrangement, suggesting that the immediate chromosomal “neighborhood” of an origin is crucial for delineating its usage. Intriguingly, these alterations in origin firing are asymmetric and do not span the inversion endpoints. Finally, our results in post-quiescent and pre-meiotic S phases are comparable, demonstrating that modifying genomic context similarly affects distinct physiological situations. Altogether, our findings provide new insight into how chromosomal architecture may shape origin selection and the program of DNA replication.

## 2. Materials and Methods

### 2.1. Strains and Growth Conditions

Fission yeast strains used in this study are listed in Appendix A. Standard *S. pombe* genetic procedures and media as in Moreno et al., 1991 were used [43]. Cells were grown in minimal medium plus supplements (EMM6S) unless otherwise indicated. To synchronize cells by nitrogen depletion, cultures were grown at 25 °C to a density of ~2 × 10^6^ cells in EMM6S. Cells were then starved of nitrogen in minimal medium without NH_4_Cl (EMM-N) for 16 h at 25 °C, leading to cell cycle exit. Release from starvation was carried out by adding a nitrogen source (0.05 g/L NH_4_Cl). In addition, leucine (0.225 g/L) and uridine (0.225 g/L) were added at the time of release to allow the optimal growth of cells carrying the *LEU2* and *ura4*^+^ markers associated with the *loxP* cassettes, as they do not fully complement the *leu1-32* and *ura4-D18* mutations. Cells were also shifted to 34 °C at this time; in strains carrying the *pat1-114* mutation [44], this results in meiotic induction. Information about the fission yeast genome sequence and annotations were obtained from PomBase [45].

### 2.2. Construction of Rearrangement Strains

To generate *S. pombe* strains with rearranged chromosomes, we used a Cre-*loxP* site-specific recombinase strategy in which recombination between *loxP* sites can be induced by transient expression of the Cre recombinase (Appendix A). For this approach, two plasmids each containing one *loxP* site were constructed. First, the *ura4* gene was cloned from pKSura4 into pFA6a-kanMx6 [46] via PmeI digestion, resulting in the generation of the pFA6a-kanMx6-ura4 plasmid (pJW1). The *loxP* sequence was then PCR amplified and cloned into pFA6a-kanMx6-ura4 via BstXI/NcoI digest, resulting in the removal of the TEF promoter of the kanMx6 cassette (pJW3). Second, the *LEU2* gene from *S. cerevisiae* was PCR amplified from pREP1 using primers containing the *loxP* sequence and integrated into pFA6a-kanMx6 via NcoI/PmeI digest, which results in the deletion of the kanMx6 ORF sequence (pJW4). The cassettes in pJW3 and pJW4 were then PCR amplified and sequentially integrated by homologous recombination in a strain auxotrophic for uracil and leucine. Integrations were selected on minimal medium plus supplements but lacking uracil or leucine, depending on the transformation, and then confirmed by PCR. Integration sites were directed by primers containing 80-bp homology to the targeted regions at the following positions: for the chromosome I inversion—3,810,766 and 4,785,016; for the chromosome II inversion—1,577,150 and 3,694,900.

To generate the chromosomal rearrangements, the strains above containing *loxP* sites integrated in the two selected loci were transformed with pJW5 for conditional expression of the Cre recombinase under the control of the *nmt41* promoter. pJW5 was generated by inserting the natMx6 cassette (EcoRV/BglII + blunt) from pFA6a-3HA-natMX6 [47] in PvuII-digested pAW5 [48]. Transformants with pJW5 were selected on EMM4S + nourseothricin (100 µg/mL) plates to induce Cre recombinase expression. A productive recombination event inserts the TEF promoter 5′ to the *kanMx6* ORF, resulting in the expression of the kan^r^ gene. Positive clones were re-streaked onto fresh EMM4S + G418 (100 µg/mL) plates. To repress Cre expression and permit loss of the plasmid carrying Cre, candidates were then grown in YE4S medium supplemented with thiamine (60 µM) for 2 days, followed by plating on YE4S + thiamine + G418. To check the loss of the plasmid expressing Cre recombinase, positive clones were re-streaked on YE4S + nourseothricin plates and screened for nourseothricin sensitivity.

For meiotic experiments, *loxP* sequences integrated at the target sites were crossed into a *pat1-114* background, and rearrangements were subsequently induced as above. Clones were selected and then diploidized using a standard protocol. Briefly, cells were grown in YE4S at 25 °C to exponential phase, treated with the microtubule-destabilizing drug carbendazim (20 µg/mL) for 5 h at 25 °C, then plated on YE4S + Phloxin B. Colonies of diploid cells appear as dark red on these plates.

### 2.3. Analysis of Cell Cycle and Meiotic Progression

For DNA content analysis by flow cytometry, cells were fixed in 70% cold ethanol, washed in 50 mM sodium citrate and treated with RNase A (0.1 mg/mL) at 37 °C overnight. Samples were then stained using 2 mg/mL propidium iodide, sonicated (Branson Digital Sonifier; Branson Ultrasonics, Brookfield, CT, USA), and analyzed using a BD Accuri C6 flow cytometer (Becton Dickinson, Franklin Lakes, NJ, USA). Analysis was performed using the FlowJo analysis software. Note that the fission yeast cell cycle differs from that of a number of model systems, as the G1 phase is short and S phase occurs prior to cytokinesis. As a result, haploid cells have a 2C DNA content during most phases of their cell cycle. The appearance of a 1C peak in nitrogen-starved cells is the result of cell cycle exit and arrest in a quiescent state in these conditions: Cytokinesis occurs without an ensuing S phase.

To assess meiotic progression, cells were ethanol fixed and stained with DAPI (2 mg/mL) to visualize nuclei. Samples were counted every 15 min to determine the kinetics of meiosis I and meiosis II: cells harboring two nuclei indicate completion of meiosis I, and four nuclei are observed upon completion of meiosis II. Microscopy was performed using an inverted Zeiss Axio Observer (Carl Zeiss Microscopy GmbH, Jena, Germany) equipped with a Lumencor Spectra X illumination system and an Orca Flash 4.0V2 sCMOS camera (Hamamatsu Photonics, Hamamatsu City, Japan). Images were acquired using the VisiView software (Visitron Systems GmbH, Puchheim, Germany) and a Plan-Apochromat 63X/1.4 NA immersion lens (Carl Zeiss Microscopy GmbH, Jena, Germany).

### 2.4. Microarray Experiments and Analyses

For origin mapping, competitive hybridizations of differentially-labelled samples were performed using Agilent 4 × 44 k *S. pombe* arrays (60-mer oligonucleotides every ~250 nucleotides; Agilent Technologies, Santa Clara, CA, USA) as previously described [42]. The copy number was determined through comparing a sample of non-replicating cells and a sample of cells undergoing DNA replication in hydroxyurea (HU). HU treatment limits the extension of DNA synthesis around the sites of initiation, allowing the identification of replication origins [5].

For our analyses, cells were starved of nitrogen for 16 h, and 12 mM HU was added when cells were released upon addition of a nitrogen source. Samples were collected at a time where bulk S phase is complete in non-HU treated cells (240 min for haploid cells undergoing a mitotic cycle; 210 min for *pat1-114* diploids). Genomic DNA was extracted [49] and purified using the Qiagen Genomic DNA kit (Genomic-tip 20/G), and samples were labeled by random priming. Briefly, 2 μg of purified genomic DNA per sample was incubated for 5 min at 95 °C with 300 μg/mL of Random Primer (481907011, Invitrogen, Waltham, MA, USA). This was followed by the addition of 2 µL Klenow (M0212S, New England BioLabs, Ipswich, MA, USA) and a nucleotide mix (0.5 mM dATP, 0.5 mM dCTP, 0.5 mM dGTP, 0.1 mM dTTP, 0.4 mM aha-dUTP) for two hours at 37 °C. aha-dUTP-labeled DNA was then purified using the PureLink PCR purification kit (K310001, ThermoFisher, Waltham, MA, USA) and ethanol precipitated using 75 mM sodium acetate and 0.1 μg/μL glycogen for 30 min at −20 °C. DNA was then washed with 70% ethanol, air dried, and resuspended in 80 μL 0.1 M sodium bicarbonate (pH 8.7). Each sample was then divided in two, and each half was incubated for 90 min in the dark with either Cy3 or Cy5 (GE Healthcare, Chicago, IL, USA) for dye swap (see below). The dye coupling reaction was stopped by the addition of 15 μL of 4 M hydroxylamine. DNA was then purified using the PureLink PCR purification kit (K310001, ThermoFisher, Waltham, MA, USA). ~1–2 μg of differentially labeled DNA from non-replicated and replicated samples were hybridized on the microarray. Two independent hybridizations of the same samples were performed in a dye-swap experiment to limit noise and dye bias. The ratios of replicated to non-replicated DNA, representing copy number, were then assessed for each hybridization, and outliers were removed prior to further analysis. For this, we considered that a ratio of 2 for replicated vs. non-replicated DNA represents 100% replication, so all values >2 were removed.

Next, for each experiment, the geometric means over five consecutive probes were determined across the genome for each reciprocally labelled dataset, and these datasets were averaged. Two biological repeats were performed for each strain, and we took the average of the data from these two experiments. For analysis and comparison of copy number, the baselines between different conditions, representing unreplicated DNA, were matched and set to 1 using a previously described approach [50]. Briefly, as cells are in HU, there are genomic regions that remain unreplicated in our S phase samples, and these will have the lowest values in each dataset. Therefore, for each averaged set of biological repeats, we took the lowest 10% of the ratios of replicated vs. unreplicated DNA and calculated their median, then normalized the dataset to this value. The following correction factors were applied: *Control I*—0.04; *Inversion I*—0.04; *Control II*—0.04; *Inversion II*—0.046; *Pat1 Control I*—0.048; *Pat1 Inversion I*—0.046; *Pat1 Control II*: 0.044; *Pat1 Inversion II*—0.037. Unless otherwise stated, the replication profiles shown in the figures are these averaged and baseline-corrected datasets, with copy numbers converted into percent efficiency (e.g., a ratio of 1.5 corresponds to 50% efficiency). Note that a consistent low outlier value at chII: 2.9838 × 10^6^ was removed from the profiles of *Control cI*, *Inversion cI*, *Inversion cII*, and *Pat1 Control cII.*

To identify origins and determine their efficiencies, we applied LOESS regression (5000 bp 2nd radius) to the averaged corrected dataset for each strain. Sites were selected as origins if the ratio of HU vs. unreplicated was greater than 1.05, representing a clear increase over the background after smoothing. Origins that were within ~5 kb were considered as a single origin and assigned the highest efficiency value. Due to the low numbers of probes in centromeric regions, these sites were not considered in our analyses; we also did not consider probes related to mating type, as it was not the same for all of our strains. Origin selection was further confirmed through visual inspection of the data. For subsequent comparisons of origin usage between *Control* and *Inversion* strains, we established a list of origins for post-quiescent S phase (201 total sites). For this, we retained origins with ratios greater than 1.05 in both the *Control* strains. Sites with ratios greater than 1.05 in either of the *Inversion* backgrounds were then added. To establish a similar list for pre-meiotic S phase (276 total sites), the same procedure was applied using the *PatI Control* and *Pat1 Inversion* datasets. Copy numbers were then converted into efficiency. These sets of origins were used to generate the profiles of replication domains in Figures 2B and 5B.

To identify origins that showed changes in usage due to the engineered chromosomal inversions, we began by establishing the level of reproducibility between the experiments we performed for the control strains. The Pearson’s correlation coefficient (r) for pairwise comparisons of origin efficiencies in *Control cI* vs. *cII* and *Pat1 Control cI* vs. *cII* showed strong positive correlations (r = 0.9884 and r = 0.9865, respectively; *p* < 0.00001 in both cases). In addition, we estimated the experimental noise in our origin mapping assays, determining the differences in origin efficiencies between the pairs of control strains. The median and median absolute deviation (MAD) of these differences show that the compared datasets are virtually identical (*Control cI* vs. *cII*: 1.6% ± 1.1%; *Pat1 Control cI* vs. *cII*: −1.5% ± 1.2%). Thus, sites with altered replication initiation due to the chromosomal inversions correspond to those whose differences in efficiency between each *Inversion*/*Control* pair are significantly beyond the variability of the method, applying a stringent cutoff of Q3 + 3IQR and Q1 − 3IQR.

To validate the analyses above, we further demonstrated the reproducibility of our methodology for determining origin efficiencies. We established that individual biological repeats for a given experiment were highly reproducible, performing pairwise comparisons of origin efficiencies (Appendix A). For this, we applied baseline correction and LOESS regression (5000 bp 2nd radius) to individual experiments. We then took the origin positions for post-quiescent and pre-meiotic S phases determined above and assigned the efficiencies at these locations using the LOESS regressions. Complementary to this, we also showed that the differences in origin usage that we observed between our chromosomal inversion strains vs. their corresponding controls are detected even with individual repeats of an experiment (Appendix A). We thus concluded that our methodology is both reproducible and robust.

## 3. Results

### 3.1. Construction and Characterization of Chromosomal Rearrangements

To test directly the impact of chromosomal context on the replication program, we generated *S. pombe* strains containing rearrangements that disrupt and fuse distinct replication domains. Specifically, we exploited a context in which we previously observed striking differences in origin efficiencies between genomic regions [42], namely cells undergoing the first S phase during cell cycle re-entry from a quiescent state. In contrast to the replication program in cycling cells, the clear separation of replication domains in this situation, as well as the dramatic differences in origin activities between domains, make it an ideal model for our study. Taking advantage of this particular profile, we constructed inversions in chromosomes I and II, where we observed the most substantial variations in origin efficiency compared to the more homogeneous profile of origin firing on chromosome III (Appendix A). To engineer the rearrangements, we used the Cre-*loxP* site-specific recombinase technology [51,52], with transient induction of Cre leading to recombination between *loxP* sites that we integrated at selected genomic sites [48] (Appendix A, see Materials and Methods). All *loxP* sites are located in intergenic regions (Appendix A), and each of the inversions contains an endpoint adjacent to a centromere (Appendix A). Importantly, the sites of *loxP* insertion were chosen to optimize our disruption strategy while minimizing potential alterations of gene expression.

First, we characterized the impact of these chromosomal rearrangements on cellular physiology by evaluating their effects on cell growth and proliferation. We determined the generation times of the inversion strains (*Inversion cI* and *Inversion cII*), comparing them to wild type as well as their parental strains prior to rearrangement (referred to as *Control cI* and *Control cII*, containing the *loxP* sites). Surprisingly, our results showed virtually identical doubling times for all tested strains (Figure 1A), demonstrating that neither the integration of *loxP* sites nor the engineered inversions affect cell growth. Next, although the rearrangements appear to have no discernable effect on overall cell proliferation, they may have more significant consequences in challenging conditions. In particular, this may be the case during the exit from the quiescent state used for our study (see above), and we thus assessed the dynamics of cell cycle re-entry in this context. To this end, cells were grown to exponential phase, depleted of nitrogen for cell cycle exit, and then induced to re-enter the cell cycle upon addition of a nitrogen source (Figure 1B). DNA content analysis by flow cytometry revealed no differences in cell cycle progression between the wild-type, *Control*, and *Inversion* strains (Figure 1C). Notably, the timings of S phase entry and the durations of S phase were comparable, despite potentially major disruptions to the replication program as a result of the rearrangements. Taken together, our data show that the alterations in genome organization induced by such large-scale chromosomal inversions do not perturb cell growth, cell cycle progression, or cell cycle re-entry from quiescence.

### 3.2. Impact of Chromosomal Inversions on the Overall Replication Program

Although we did not observe differences in S phase progression in strains harboring the engineered inversions, the impact of these rearrangements on the replication program may be significant. For instance, fission yeast cells lacking the Rif1 protein, a regulator of DNA replication, show no defects in cell growth and proliferation but display major, genome-wide alterations in origin usage [53]. We therefore set out to determine the program of DNA replication along the chromosomes in our models, in which a large number of origins are relocated to different chromosomal coordinates. For these experiments, cells were synchronized in quiescence using the nitrogen depletion procedure as above and allowed to resume the cell cycle and enter S phase in the presence of 12 mM hydroxyurea (HU). The use of HU, which limits the extent of DNA synthesis surrounding sites of initiation, is a well-established method for identifying origins and ascertaining their efficiencies [5,6]. The replication profile was assessed by competitive hybridization of differentially-labelled unreplicated and S phase samples to microarrays [50,54]. A detailed explanation of the methodology for origin identification and efficiency analysis is provided in the Materials and Methods section.

We began by evaluating the overall replication program in *Inversion cI*, which contains the smaller of the two rearrangements, comparing it with that of *Control cI*. This inversion of a ~1 Mb genomic region, which is bounded by an efficient domain at one end and an inefficient region on the other, juxtaposes genomic “neighborhoods” that normally possess contrasting origin activities (Appendix A). As shown in Figure 2A (left panel), our results reveal the major disruption of an efficient domain, along with the appearance of a new high-efficiency cluster. This substantial re-organization of replication is further highlighted by the regional profiles of origin usage generated using these data (Figure 2B, left panel). Next, we asked whether the larger rearrangement in *Inversion cII* results in similar effects on replication initiation using the same approach. We found that this ~2 Mb inversion, which spans almost half of chromosome II, also induces extensive alterations in the organization of DNA replication (Figure 2A,B, right panels). Efficiency domains are both fractured and created, as observed for *Inversion cI*. Interestingly, the changes in the replication programs of the *Inversion* strains are restricted to the rearranged chromosomes, indicating that these modulations do not have secondary effects for DNA synthesis in *trans*. Altogether, our results demonstrate that large genomic inversions in fission yeast lead to a restructuring of replication initiation along the chromosomes and that chromosomal position does not dictate the characteristics of replication of a given genomic region.

### 3.3. Consequences of Chromosomal Inversions for Origin Selection

As our analyses suggest major changes in replication domains due to chromosomal inversions, we next ascertained the effects at the level of individual origins. Indeed, the observed alterations may result from widespread modifications of origin usage or from relocation of initiation sites across the inverted region. To facilitate the direct comparison of origin activities in the *Control* and *Inversion* strains, we aligned the replication data of the different genetic backgrounds by plotting those of the *Inversions* using the original, pre-translocation coordinates. Note that this representation contrasts with the graphs in Figure 2, where the data were shown with their actual post-inversion positions along the chromosomes. Remarkably, we found that the efficiencies of origins are largely unaffected (Appendix A). Indeed, for both *Inversions*, our results demonstrate that there is no effect of re-organizing the genome on overall origin activities, with the medians and median absolute deviations of the differences in the efficiencies of all origins between each rearrangement and its corresponding control being minimal (*Inversion cI*: 1% ± 1%; *Inversion cII*: 1.5% ± 0.9%). These differences are similar to the experimental variability determined using the two *Control* backgrounds (1.6% ± 1.1%). Collectively, these data indicate that the global effect described above is brought about by relocating origins that maintain their efficiencies.

However, when focusing on the origins that are most exposed to a new genomic neighborhood, i.e., those in close vicinity of the inversion endpoints, we found striking differences in replication initiation events. In *Inversion cI*, we observed changes within a distance of ~100 kb of the disruptions in genomic context (Figure 3A). All efficient origins within the inverted fragment that are now located adjacent to an origin-poor domain lose their activity (Figure 3A, Top left; decreases of 22.9%, 10.1%, and 14.6%). Complementary to this, origins that normally display little or no activity show strong efficiency increases (+9.6% and +33.2%) when translocated next to an efficient domain (Figure 3A, Top right). Interestingly, the changes in origin firing only occur within the rearranged fragment, and no alterations are detected in external regions (Figure 3A, bottom). Next, in *Inversion cII*, a considerably larger rearrangement, specific modifications in activity were also identified (Figure 3B). In this case, differences were observed at all 6 origins that are located within ~150 kb of the inversion endpoints. However, in contrast to *Inversion cI*, these were found strictly external to the rearranged fragment. These alterations in origin selection between the *Inversion* and *Control* backgrounds were the only ones greater than our significance threshold (Figure 3C), and none were present in the parental *Control* strains (Appendix A). These results therefore provide evidence that the changes in origin efficiency are indeed due to perturbations in genomic context rather than the insertion of the *loxP* cassettes. Moreover, there is a correlation between the amplitude of the differences in origin usage and the distance from the inversion endpoints (Figure 3A,B, bottom), suggesting a positional effect of disrupting the genomic environment.

Taken together, our analyses indicate that large chromosomal rearrangements do not modify the activities of individual origins across the genome except in regions that adjoin new chromosomal neighborhoods. Surprisingly, the changes in origin efficiency that arise as a result of these inversions occur either inside (*Inversion cI)* or outside (*Inversion cII*) of the rearranged regions but do not appear to span the transition points. This “asymmetric” effect hints at the possibility that the architecture of longer fragments is imposed upon bordering sequences, while a smaller inversion is subject to regulation by the characteristics of its surrounding regions. Our findings thus suggest that while changes in the positions of initiation sites, rather than their activities, are responsible for the overall reorganization of the replication program brought about by gross chromosomal rearrangements, origins near the fusion points are exquisitely sensitive to their new genomic contexts.

### 3.4. Consequences of Chromosomal Inversions for the Meiotic DNA Replication Program

Given the impact of the chromosome I and II inversions on the replication program in our post-quiescence model, we asked if they might have more consequential effects in a physiological context that involves a distinct chromosomal biology. The process of meiosis is a specialized reductional division with specific events in DNA metabolism, in particular interhomolog recombination and two rounds of chromosome segregation [55]. Meiosis in fission yeast is characterized by the arrangement of chromosomes in a “bouquet” in which telomeres are clustered, with an elongated nucleus that moves back and forth between the cell ends during meiotic prophase [56]. These complex mechanisms may thus make meiosis more sensitive to large-scale changes in genomic context and in the organization of DNA replication. We therefore constructed homozygous diploid strains that combine the *pat1-114* temperature-sensitive mutation, which allows for meiotic induction after temperature shift [57], with either the *Control* or *Inversion* backgrounds. First, we characterized the phenotypes of these strains, beginning with their generation times at 25 °C, the permissive temperature for *pat1-114*. Our data showed that chromosome I and II inversions do not affect the proliferative growth of *pat1-114* diploid cells (Figure 4A). Importantly, we then analyzed whether these rearrangements lead to defects in meiotic progression. For these experiments, we used the same protocol as above: *pat1-114* diploid cells were grown to exponential phase, depleted of nitrogen, then supplemented with a nitrogen source and shifted to a non-permissive temperature to trigger synchronous meiotic entry. DNA content analysis showed that pre-meiotic S phase takes place at the same time and with similar durations in all strains (Figure 4B). We then assessed the timing and execution of meiosis I and II following the completion of DNA replication (Figure 4C). Unexpectedly, our results showed no detectable changes in meiotic progression in the presence of either chromosomal rearrangement. Altogether, we conclude that despite the complex behavior of meiotic chromosomes in fission yeast, large genomic rearrangements do not perturb meiotic entry or progression.

Next, we asked whether alterations in chromosomal context might have distinct and amplified effects on the replication program of pre-meiotic S phase. To this end, we determined the origin firing profiles of the diploid rearranged *pat1-114* strains (*Pat1 Inversion cI* and *Pat1 Inversion II*) as well as that of their respective controls (*Pat1 Control cI* and *Pat1 Control cII*); see Materials and Methods for detailed protocols. Consistent with our previous observations [42], the organization of DNA replication in the *Pat1 Control* backgrounds is very similar to that in cells exiting quiescence (compare *Control* vs. *Pat1 Control* profiles in Appendix A; also compare Figure 2 with Figure 5). Strikingly, our results demonstrate that chromosomal rearrangements have the same effects in both post-quiescent and pre-meiotic S phases. First, the global reorganization of DNA replication in the *Pat1 Inversion* backgrounds (Figure 5) is solely brought about by changes in origin positioning along the rearranged chromosomes. Indeed, similar to our findings for post-quiescent S phase, the activities of individual origins are maintained regardless of their new locations: the median and median absolute deviation of the differences in efficiencies of all origins between each inversion and its control are minimal (*Pat1 Inversion cI* vs. *Pat1 Control cI:* 0.4% ± 1.5%; *Pat1 Inversion cII* vs. *Pat1 Control cII*: 0.2% ± 1.1%; compare with *Pat1 Control cI* vs. *Pat1 Control cII:* −1.5% ± 1.2%). This indicates that major alterations in chromosomal organization do not have more pronounced effects on sites of replication initiation, despite the signature behavior of meiotic chromosomes. In addition, a small subset of origins proximal to the inversion endpoints have significantly altered efficiencies (Figure 6). These include all of the sites identified in our post-quiescent S phase assays, and we detected additional nearby origins in meiotic cells whose activities are likewise affected (compare Figure 6A,B with Figure 3A,B). Moreover, the asymmetry and position effect of these changes is identical to what we observed (Figure 6A,B, bottom).

Collectively, these observations demonstrate that the genomic neighborhood of an origin, rather than its chromosomal position, is a pivotal determinant of its activity in both post-quiescence and pre-meiotic S phase. Our study therefore reveals that even in cellular states with considerably different behaviors in chromosomal and chromatin biology, genomic restructuring gives rise to common effects on the overall organization of DNA replication and the selection of individual origins.

## 4. Discussion

The organization of DNA replication in chromosomal domains is a conserved feature of eukaryotic genome architecture, but we still do not understand key aspects of how the replication program is modulated. In this study, we have probed the impact of genomic context on origin selection, exploiting models that combine engineered chromosomal rearrangements with the unique replication programs of post-quiescent and pre-meiotic S phase. Our results demonstrate that inversions of extended regions within chromosomes I and II do not have detectable effects on cellular physiology, whether for cell growth and proliferation or for meiotic progression. Remarkably, while disruptions to and emergence of replication domains were observed along each rearranged chromosome, the efficiencies of individual origins remained virtually unaffected genome wide. This indicates that origin repositioning is the principal cause of the reprogramming of DNA replication that occurs in these conditions. However, characteristic changes in origin activity arose near the fusion points of each inversion, implicating the immediate chromosomal neighborhood of an origin as a critical determinant of its usage. Interestingly, all of these effects are virtually identical in both post-quiescent and pre-meiotic S phases, suggesting that despite notable differences in chromosomal behavior, similar principles govern the regulation of origin selection by genomic context in these physiological states.

One of the most striking observations from our work is that large-scale chromosomal inversions induce localized effects on replication initiation. This targeted impact of disrupting genomic context that we uncovered may be a major contributor to the changes in replication timing that were described near rearrangement endpoints of syntenic regions in different species [40,41]. The local alterations in origin usage that we identified possess several intriguing characteristics. First, although we may have expected an “equalizing” effect of fusing regions with vastly different efficiencies, leading to increases and decreases on either side of a breakpoint, this is not the case. Indeed, changes are surprisingly found on only one side of each endpoint, either within (chromosome I) or outside of (chromosome II) the inverted fragment. Moreover, origins distal to the inversion breakpoints remain unaffected. This latter finding is consistent with earlier work that revealed the overall conservation of replication timing for a highly rearranged human chromosome 21 introduced in mouse cells, with the majority of differences in timing arising near rearrangement points [39]. Given our results and the links between the organization of DNA replication and chromosome topology [15,35,36,58,59,60,61], we speculate that the specific modulations of origin usage in our models are brought about by a perturbation of chromosomal contacts proximal to the rearrangement endpoints. Following an inversion, the genomic region on one side of the breakpoint may impose its architecture on the other side. In this case, the alterations that we observe would effectively appear to be asymmetric and not to span the points where genomic context is disrupted, as origins on the side that does not change conformation would maintain their firing properties. This may also explain why initiation sites distal to the inversion breakpoints remain unaffected: the further away they are, the less likely they are to be influenced. However, the parameters that determine which side of an inversion dictates the context of a fusion locus remain unclear. While our analyses and previous studies [39] hint at the idea that longer fragments may constrain the architectures of adjacent regions, this may represent only one of the factors involved in this regulation. Indeed, ERCEs remain early-replicating even when placed in ectopic genomic environments through chromosomal inversions [36]. Altogether, our findings suggest that the immediate genomic neighborhood of a locus may impinge on its chromosomal topology and thereby modulate local origin firing.

How, then, might chromosomal inversions bring about changes in architecture that result in alterations in the replication program? In fission yeast, the genome is organized in small (50–100 kb) and large (100 kb–1 Mb) domains that are mediated by cohesin and condensin, respectively [62,63]. Chromosomal architecture changes during the cell cycle, as small chromatin domains are observed in interphase, while fewer but larger domains are set up during mitosis [64,65]. Interestingly, the centromeres of the three chromosomes are clustered, with interactions frequently observed between centromere-proximal regions [62,66]. Our engineered chromosomal inversions, both of which have one endpoint near a centromere, may thus disrupt or create new contacts that organize genome conformation in these areas. Similarly, telomeres have been shown to associate with chromosomal arm regions that contain late-replicating origins bound by Taz1, a component of the Shelterin telomere protection complex [67]. Alteration of the positions of these sites in the genome as a result of our engineered inversions may therefore modulate the replication timing of nearby origins. An understanding of the changes in chromosomal topology induced by genome rearrangements may thus contribute to elucidating how local and large-scale alterations in chromosomal architecture may shape the spatiotemporal program of DNA replication.

On the other hand, our findings also highlight the robustness of the mechanisms that determine origin activity. Although numerous origins change their positions following large-scale chromosomal inversions, their firing appears to be remarkably unaffected, as the efficiencies of individual sites are mostly preserved across the genome. This robustness may provide an explanation for the similarities in replication timing observed for syntenic regions of different species even in the presence of multiple rearrangements [39,40,41]. Perhaps paradoxically, the maintenance of origin activities, together with the extensive changes in origin coordinates in our models, naturally gives rise to a complete re-organization of replication domains along the chromosomes. Unexpectedly, these significant alterations do not appear to have deleterious effects for cellular physiology. Nevertheless, while we have evaluated the overall impacts on cell proliferation and meiotic progression, there may be more specific consequences for processes related to DNA metabolism, such as gene expression, DNA repair, and recombination [68,69]. Indeed, we have previously demonstrated that origin activity in pre-meiotic S phase contributes to the regulation of meiotic recombination [42], hinting at the possibility that the re-organization of DNA replication in our models may have ramifications for the exchange of genetic material. Furthermore, although genome rearrangements may not bring about obvious short-term defects, there may be important long-term consequences. For instance, there may be effects on cellular fitness that are not discernible in our experimental assays but are critical for long-term survival or for responding to challenging environments. In addition, the coupling of replication timing with mutation frequency and spectrum [70,71,72,73] suggests potential repercussions for cellular adaptation and evolution. Related to these ideas, genome rearrangements in evolution appear to be enriched for fusions of regions that replicate with the same timing characteristics, which may suggest a selection for non-deleterious disruptions [40].

Taken together, our study represents an important step in our understanding of the coupling between DNA replication and chromosomal architecture. The investigation of this intricate interplay will provide crucial insight into the remarkable robustness and flexibility of the replication program, with implications for the conservation of this feature of genome organization throughout eukaryotic evolution.

## Figures and Tables

**Figure 1 genes-13-01244-f001:**
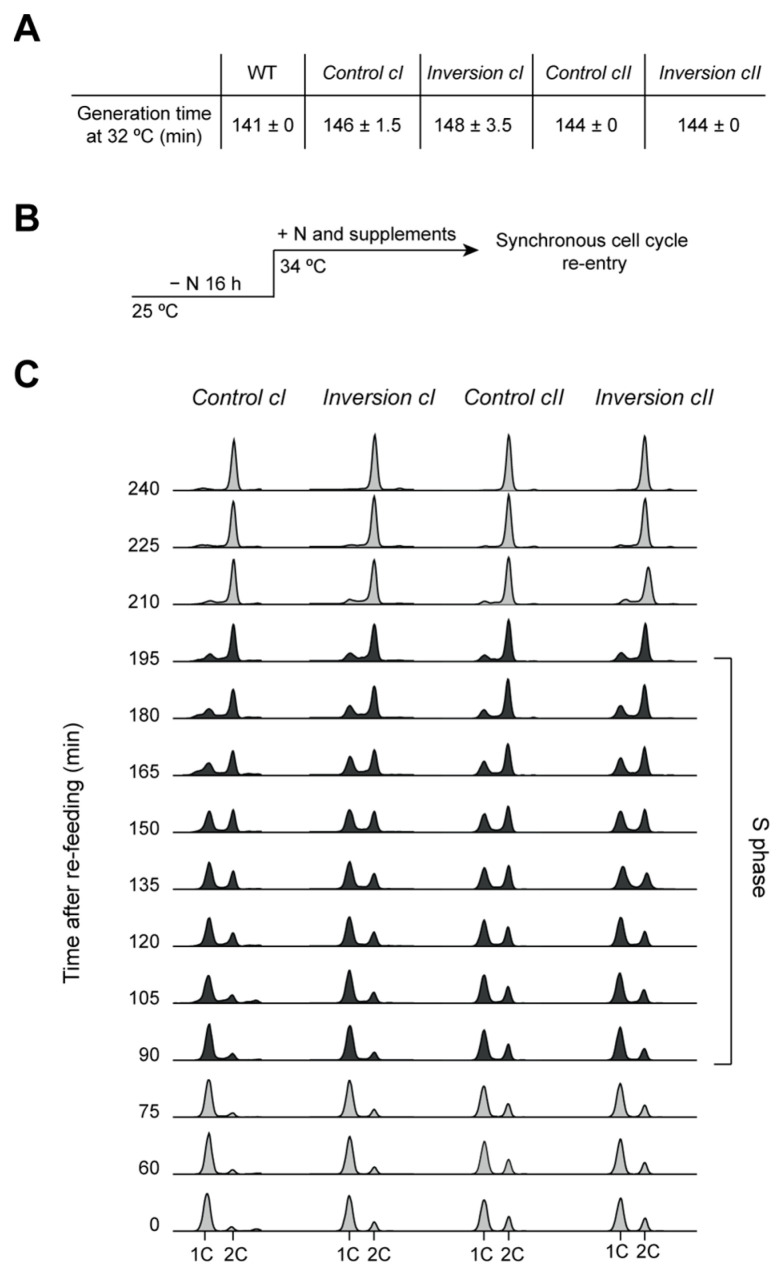
Characterization of cells harboring chromosomal inversions. (**A**) The generation times of the two inversion strains, along with the two parental strains, are virtually identical to that of wild-type (WT) cells. Cells were grown in EMM6S at 32 °C. Averages and standard errors of two independent experiments are shown. (**B**) Experimental design for nitrogen depletion and cell cycle exit, followed by synchronous re-entry into a proliferative cycle. Cells were grown to exponential phase in EMM6S, starved of nitrogen at 25 °C for 16 h, and induced to re-enter the cell cycle upon addition of NH_4_Cl. To allow for comparisons with pre-meiotic S phase and meiotic progression in the *pat1-114* background (see Figure 4), cultures were shifted to 34 °C, the restrictive temperature for *pat1-114,* at the time of NH_4_Cl addition. (**C**) DNA content analysis of the first S phase during cell cycle re-entry from quiescence. NH_4_Cl was added to the medium at T = 0. Time points during which cells undergo bulk S phase are indicated in black. S phase onset and duration are identical in the *Control* and *Inversion* strains.

**Figure 2 genes-13-01244-f002:**
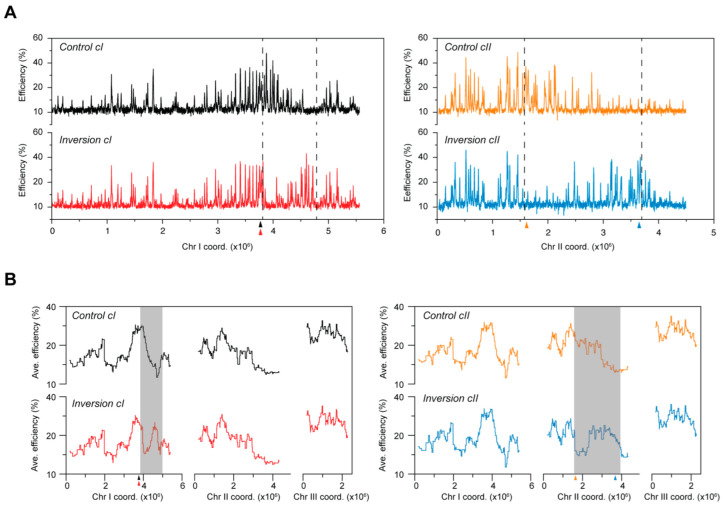
Alterations in replication domains resulting from chromosomal inversions. (**A**) Replication profiles of the chromosomal inversions plotted using the actual post-rearrangement chromosomal coordinates. Only the chromosomes harboring the inversions are shown. Black: *Control cI*, red: *Inversion cI*, orange: *Control cII*, blue: *Inversion cII.*
*x*-axis: chromosome coordinates, *y*-axis: origin efficiencies. (**B**) Profiles of replication domains. Averages of origin efficiencies were determined for continuous windows of ~250 kb (1000 probes) across the entire genome. *x*-axis: chromosome coordinates, *y*-axis: average origin efficiencies. Gray boxes highlight changes in origin efficiency domains along the chromosomes that result from chromosomal inversions. Triangles with colors corresponding with each of the strains indicate the positions of centromeres.

**Figure 3 genes-13-01244-f003:**
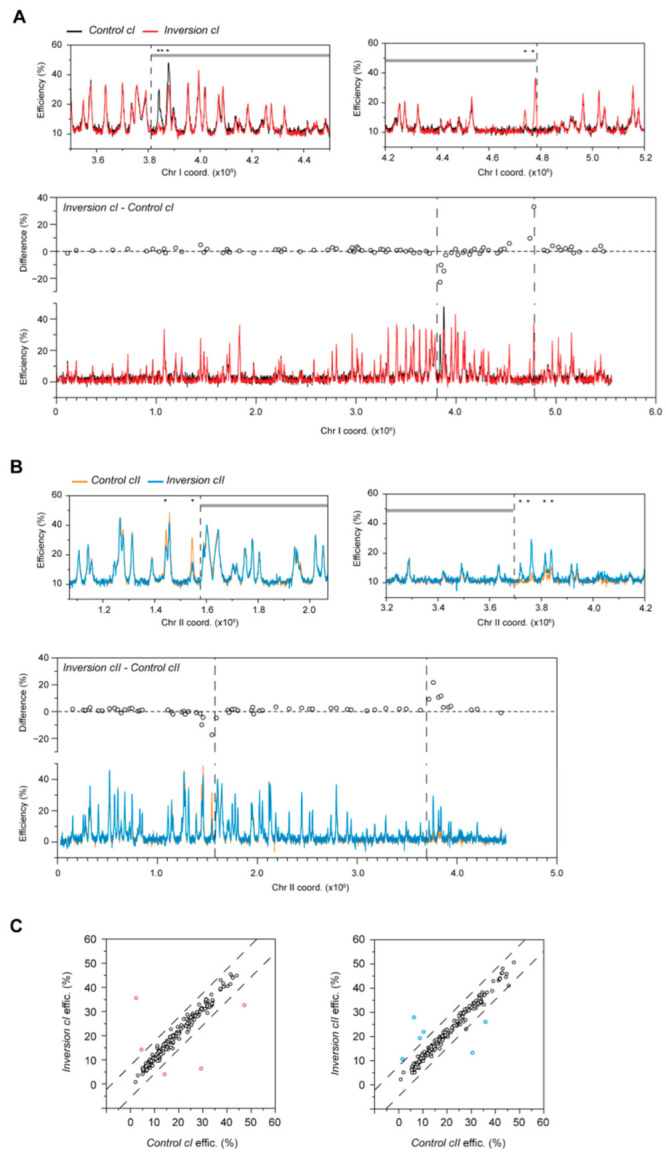
Alterations in the efficiencies of individual origins due to chromosomal inversions in haploid fission yeast. (**A**,**B**) Differences in replication initiation between *Inversion* and *Control* strains. To allow direct comparison with the *Control* strains, the profiles of *Inversion cI* and *Inversion cII* are plotted using the original, pre-translocation coordinates. Only chromosomes harboring the inversions are shown. Black: *Control cI*, red: *Inversion cI*, orange: *Control cII*, blue: *Inversion cII*. Dashed lines indicate rearrangement endpoints. *x*-axis: chromosome coordinates; *y*-axis: origin efficiencies. Top panels: Detailed view of the replication profiles surrounding the inversion endpoints. Alterations in origin usage in *Inversion cI* occur within the inverted fragment, whereas changes in *Inversion cII* are located outside of the inversion. Asterisks highlight origins that show significant efficiency differences between the *Inversion* and *Control* backgrounds. Gray lines delineate the rearranged fragment. Bottom panels: Position-effect of the impact of chromosomal context on origin firing. Differences in origin efficiencies between *Inversion* and *Control* strains (circles, top part of each graph) in post-quiescent S phase. A: *chromosome I* inversion; B: *chromosome II* inversion. The bottom parts of the graphs are as in Appendix A. (**C**) Comparison of origin efficiencies in the *Inversion* vs. *Control* strains. Left: *chromosome I* inversion; right: *chromosome II* inversion. Each point represents an origin. Dashed lines indicate Q3 + 3IQR and Q1 − 3IQR for the differences in origin efficiency between the compared strains. Points in red (*Inversion cI*) and blue (*Inversion cII*) are those that fall outside of these thresholds and correspond with the changes indicated in (**A**,**B**).

**Figure 4 genes-13-01244-f004:**
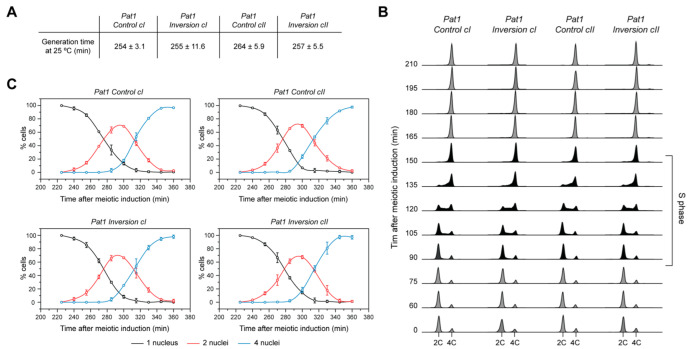
Characterization of meiotic entry and progression in strains with chromosomal rearrangements. (**A**) Generation times of diploid *pat1-114* strains with chromosome I and II inversions, along with their pre-rearrangement controls. Averages and standard errors of three independent experiments are shown. (**B**) Flow cytometry analysis of cells undergoing meiosis. Cells were depleted of nitrogen for 16 h, then shifted to 34 °C upon addition of NH_4_Cl to induce meiotic entry (T = 0). Time points during which cells undergo bulk pre-meiotic S phase are indicated in black. S phase onset and duration are identical in the different strains. (**C**) Time course of meiotic progression. Cells were ethanol fixed and stained with DAPI at the indicated time points to assess the number of nuclei. Graphs represent the average of two independent experiments (n > 300 for each time point; error bars: standard deviations). Cells with two nuclei have completed meiosis I, while those with 4 nuclei have completed meiosis II. No changes in meiotic progression are detected in the strains harboring chromosomal inversions.

**Figure 5 genes-13-01244-f005:**
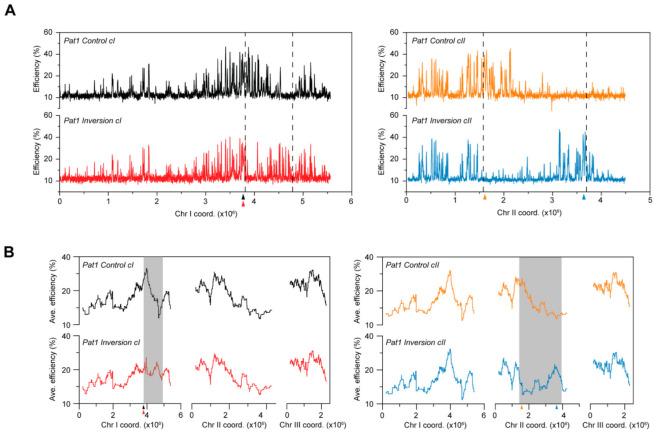
The impact of chromosomal inversions on the replication program of pre-meiotic S phase. (**A**) Replication profiles of the chromosomal inversion strains during pre-meiotic S phase plotted using the actual post-rearrangement chromosomal coordinates. Only the rearranged chromosomes are shown. Black: *Pat1 Control cI*, red: *Pat1 Inversion cI*, orange: *Pat1 Control cII*, blue: *Pat1 Inversion cII*. *x*-axis: chromosome coordinates, *y*-axis: origin efficiencies. (**B**) Profiles of replication domains. Averages of origin efficiencies were determined for continuous windows of ~250 kb (1000 probes) across the entire genome. *x*-axis: chromosome coordinates, *y*-axis: average origin efficiencies. Gray boxes highlight changes in origin efficiency domains along the chromosomes that result from chromosomal inversions. Triangles with colors corresponding with each of the strains indicate the positions of centromeres.

**Figure 6 genes-13-01244-f006:**
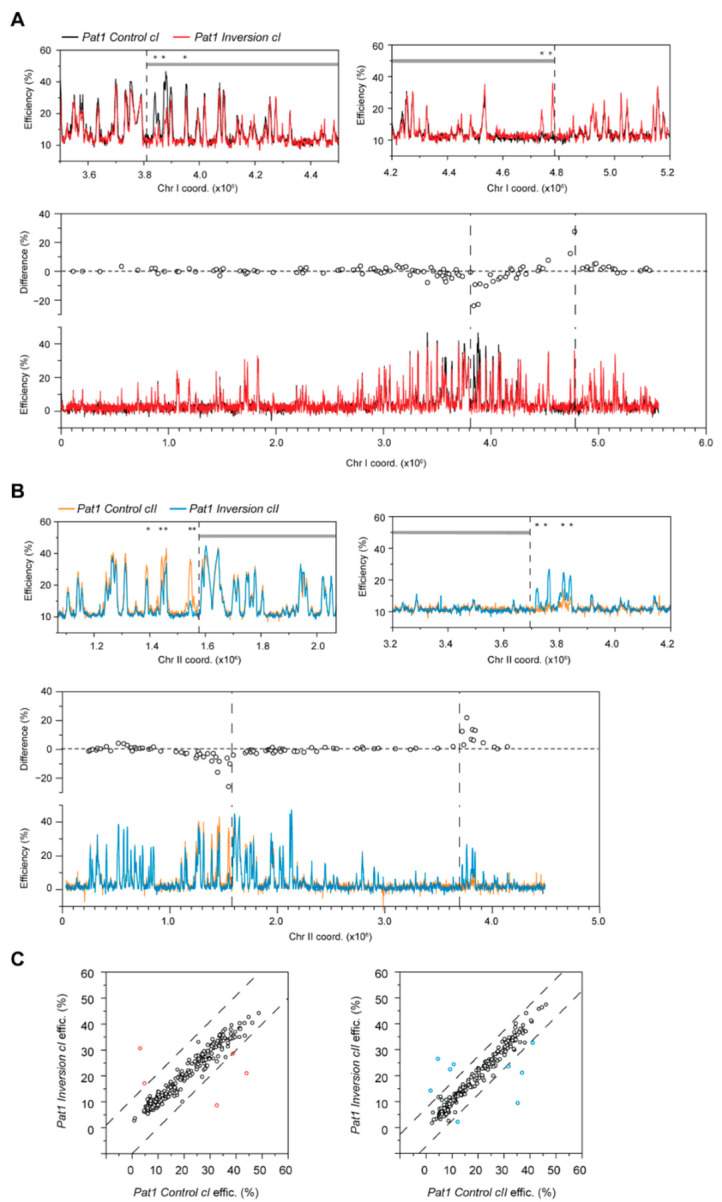
The impact of chromosomal inversions on the efficiencies of individual origins in pre-meiotic S phase. (**A**,**B**) Differences in replication initiation between inversion and control strains in pre-meiotic S phase. To allow direct comparison with the *Pat1 Control* strains, the profiles of *Pat1 Inversion cI* and *Pat1 Inversion cII* are plotted using the original, pre-translocation coordinates. Only chromosomes harboring the inversions are shown. Dashed lines indicate rearrangement endpoints. *x*-axis: chromosome coordinates; *y*-axis: origin efficiencies. Top panels: Detailed view of the replication profiles surrounding the inversion endpoints. Changes in origin usage in *Pat1 Inversion cI* occur within the inverted fragment, whereas changes in *Pat1 Inversion cII* are located outside of the inversion. Black: *Pat1 Control cI*, red: *Pat1 Inversion cI*, orange: *Pat1 Control cII*, blue: *Pat1 Inversion cII*. Asterisks highlight origins that show significant efficiency differences between the *Inversion* and *Control* backgrounds. Gray lines delineate the rearranged fragment. Bottom panels: Position-effect of the impact of chromosomal context on origin firing in pre-meiotic S phase. Differences in origin efficiencies between *Pat1 Inversion* and *Pat1 Control* strains (circles, top part of each graph). Chromosome I inversion; B: chromosome II inversion. The bottom parts of the graphs are as in Appendix A. (**C**) Comparison of origin efficiencies in the *Pat1 Inversion* vs. *Pat1 Control* strains. Left: chromosome I inversion; right: chromosome II inversion. Each point represents an origin. Dashed lines indicate Q3 + 3IQR and Q1 − 3IQR for the differences in origin efficiency between the compared strains. Points in red (*Pat1 Inversion cI)* and blue (*Pat1 Inversion cII)* are those that fall outside of these thresholds and correspond with the changes indicated in (**A**,**B**).

## Data Availability

The array data reported in this paper are available in the NIH GEO database.

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
