# Peer review of "Impact of Chromosomal Context on Origin Selection and the Replication Program"

_genes, 2022, doi:10.3390/genes13071244_

Round 1

Reviewer 1 Report

The manuscript entitled ‘Impact of chromosomal context on origin selection and the replication program’ by L. Lanteri et al explores the impact of chromosomal reorganization (using genomic rearrangments) on replication activation and replication timing in post-quiescence and premeiotic S. pombe cells.

The manuscript is well written and describes high quality experimental data.

Mayor comments:

The authors find that chromosomal rearrangments only impact the replication programing around the corresponding integration sites. Accordingly, they suggest that the chromatin context (or chromosomal neighborhood) is a major determinant of replication origin activity modulation. The manuscript would improve if the authors would include a bit more information or comment on structural features (for example genes, centromeres,…..) present at the inversion sites that have been generated in their experimental model.

Minor comments:

Figures 1C and 4B: I would suggest to swap the order of FACS analysis to ‘Control CI, Inversion CI, Control CII, Inversion CII.

Figure 4C: Numbering of the X axis should be revised.

Reviewer 2 Report

In this paper the authors look at the effects on mitotic and meiotic replication of inversion of regions of 2 chromosomes in S.pombe.

The data is very clearly presented and the paper is well written, but at this point seems to me to be a little preliminary.

Specific comments

Figure 2

I am not convinced that the way the data is presented in figure 2 shows very much. A more detailed discussion of the region that is inverted is shown in fIgure 3 and the regions outside of the inverted regions are not in enough detail to see small differences in patterns. I think seeing the regions well away from the inversions in detail is useful because it will give some indication of how much variability there is between different repeat runs. The methods section suggests that the experiments are only carried out twice and some the variations that they observe for Figure 3 are not that large.

Figure 3

I think for this it would be useful to have a more quantitative idea of the effects seen and how much they varied between different runs. I am also interested to know how much the traces were affected by the introduction of the lox sites ( ie a comparison with the wild type). I think that averaging two runs is only useful if the repeats show very little variation. Without this kind of information it is quite hard to know how strong the data is.

Similar comments apply to figures 4 and 5.

I think the paper would be significantly improved by the inclusion of some mechanistic analysis to back up and strengthen the changes observed, which by themselves are relatively subtle.  For instance:

You might expect that some of these changes would be accompanied by changes in chromatin modification if they have switched between more and less active regions. I think that at least an attempt should be made to look at this. Histone modification antibodies are relatively available for ChIP and as the sequence of the region is known it would be relatively easy to design primers to look at the relevant regions. At least in this case it would go someway to looking at the mechanisms involved in this rather than just reporting an observation.

It might also be possible to use MNase digestion to tell if the regions have changed in their degree of open-ness.

The authors speculate that the changes are due to alterations in contact points between the sites. This could be determined using chromosome capture techniques.

Round 2

Reviewer 2 Report

My only remaining comment is that in Figure 3c the different panels are side by side not top and bottom.